



# Ice speed of a Greenlandic tidewater glacier modulated by tide, melt, and rain

Shin Sugiyama[1,2] Shun Tsutaki[3,4], Daiki Sakakibara[1,2], Izumi Asaji[1], Ken Kondo[1], Yefan Wang[1], Evgeny Podolskiy[2], Guillaume Jouvet[5,6] and Martin Funk[6]

[1] Institute of Low Temperature Science, Hokkaido University, Sapporo, Japan

[2] Arctic Research Center, Hokkaido University, Sapporo, Japan

[3] National Institute of Polar Research, Tokyo, Japan

[4] Department of Polar Science, The Graduate University of Advanced Studies, Tokyo, Japan

[5] Institute of Earth Surface Dynamics, University of Lausanne, Lausanne, Switzerland

[6] Laboratory of Hydraulics, Hydrology and Glaciology, ETH Zurich, Zurich, Switzerland

*Correspondence to*: Shin Sugiyama (sugishin@lowtem.hokudai.ac.jp)

**Abstract.** Ice discharge from the Greenland ice sheet is controlled by tidewater glacier flow speed, which shows significant variations in different timescales. Short-term speed variations are key to understanding the physical processes controlling glacial motion, but studies are sparse for Greenlandic tidewater glaciers, particularly near the calving front. Here, we present high-frequency ice speed measurements performed at 0.5–4 km from the front of Bowdoin Glacier, a tidewater glacier in

northwestern Greenland. Three GPS (global positioning system) receivers were operated for several weeks in July of 2013–2017 and 2019. Horizontal ice speed varied over timescales of hours to days, including short-term speed-up events as well as diurnal and semidiurnal variations. Frequency analysis revealed that semidiurnal signals decay upglacier, whereas diurnal signals are consistently observed over the area of study. Speed-up events were associated with heavy rain, and longer-term variations were correlated with air temperature. Uplift of the glacier surface was observed during fast-flowing periods,

suggesting basal separation due to elevated water pressure. These observations confirm the strong and immediate impact of melt/rainwater on subglacial water pressure and sliding speed. Tidally modulated ice speed peaks coincided with or slightly before low tide, which demonstrates the key role viscoelastic ice dynamics play in response to changing hydrostatic pressure acting on the glacier front. Our study results reveal details of short-term flow variations near the front of a Greenlandic tidewater glacier and provide insights into calving glacier dynamics. During melt season, ice speed is controlled by atmospheric

conditions through meltwater production and rain events as commonly observed in alpine glaciers, but additional complexity arises from tidal influence near the calving front.

## 1 Introduction

Ice flow speed near the front of outlet glaciers is critically important for mass change in the Greenland ice sheet. The ice speed directly controls ice discharge from glaciers into the ocean, which accounts for approximately half of the ice sheet's ablation

(e.g. Mouginot et al., 2019). In fact, rapid mass loss of the Greenland ice sheet in the 21st century is attributed to the acceleration of marine-terminating outlet glaciers, as well as increasingly negative surface mass balance (The IMBIE Team,



2020; Mouginot et al., 2019; van den Broeke et al., 2009). Ice dynamics near the calving front are also crucial for iceberg calving (e.g. Benn et al., 2007). Therefore, studying glacier dynamics near the ocean boundary is key to understanding the current and future mass loss of the ice sheet.

In addition to changes in flow dynamics on annual or greater temporal scales, glaciers show shorter-term ice speed variations over hours to weeks. For example, glaciers accelerate when basal sliding is enhanced by subglacial pressure elevated by meltwater input to the base (e.g. Bindschadler, 1983). The impact of the pressure perturbation on the sliding is more significant, particularly near the front of calving glaciers, where subglacial pressure is consistently high close to the ice overburden level (Kamb et al., 1994; Sugiyama et al., 2011). Moreover, the dynamics of marine-terminating glaciers are affected by tides

because the hydrostatic pressure acting on the ice front constitutes a significant contribution to the glacier force budget. Short-term ice speed variations and driving mechanisms provide an important insight into the dynamics of calving glaciers, including rapidly changing outlet glaciers in Greenland.

Although satellite observations are available in increasingly high spatial and temporal resolutions, studying glacier speed variations on the time scale of hours to days requires ground-based observations. Because in-situ measurement is difficult for

logistical reasons, short-term flow variations near the calving front have been reported in a limited number of glaciers in the world. In Greenland, this list includes: Heilheim Glacier, Jakobshavn Isbræ (Sermeq Kujalleq), and Bowdoin Glacier (Kangerluarsuup Sermia). GPS (global positioning system) measurements at Heilheim Glacier within 12 km from the glacier front showed tidal modulation of the ice speed (de Juan et al., 2010). The glacier flowed faster during the falling phase of semidiurnal tides with peak speeds occurring several hours before the lowest tide and magnitude of the speed change

decreasing upglacier (Voytenko et al., 2015). Further investigations of the GPS data demonstrated short-term flow variations driven by surface meltwater draining into the bed, including diurnal speed variations (Andersen et al., 2010; 2011; Davis et al., 2014; Stevens et al., 2021). Ice speed variations as a response to tidal and meltwater forcings have been observed at Jakobshavn Isbræ as well. GPS and terrestrial radar measurements showed that speed was anticorrelated with tidal height (fast flow during low tide) (Podrasky et al., 2014; Xie et al., 2018). In addition, diurnal and daily speed variations were seen and

attributed to surface meltwater production (Podrasky et al., 2012). Near the front of Bowdoin Glacier in northwestern Greenland, ice speed was affected by tide, temperature, and rain (Sugiyama et al., 2015). Tide-modulated flow variations induced semidiurnal seismic activities on the glacier, by changing the strain rate along the glacier (Podolskiy et al., 2016).

The above-mentioned studies on the three Greenlandic glaciers have shown tidal and meltwater influences on the ice dynamics as observed at tidewater glaciers in Alaska (e.g. Walters and Dunlap, 1987; O'Neel et al., 2001). General features of the tide-

induced flow variations observed in Greenland broadly agree with those from Alaskan glaciers, in terms of the timing of variations and the upglacier extent of the tidal influence. Glaciers flow faster when tidal height is low, suggesting the hydrostatic ocean water pressure is the primary control of speed change (Walters and Dunlap, 1987; Voytenko et al., 2015; Sugiyama et al., 2015). The magnitude of the speed change rapidly decreases upglacier within a distance of several to ten kilometers (O'Neel et al., 2001; de Juan et al., 2010; Sugiyama et al., 2015). These features differ significantly from those

reported for Antarctic glaciers, where peak ice speed typically coincides with maxima in falling tidal velocity rather than the



lowest tide (Anandakrishnan et al., 2003; Marsh et al., 2013) and tidal influence extends several tens of kilometers inland from the grounding line (Bindschadler et al., 2003; Gudmundsson., 2006; Murray et al., 2007). Interestingly, measurements at Nioghalvfjerdsbr, one of a few Greenlandic glaciers having large floating ice, showed little tidal influence on the flow speed above the grounding line (Christmann et al., 2021). Therefore, processes driving the tidal modulations are likely different in

Antarctica and other regions (Thomas, 2007). The increasing number of measurements in Greenland indicates that surface meltwater penetrates to the glacier base, affecting the subglacial conditions and basal sliding speed. This is consistent with observations at temperate glaciers, including tidewater glaciers in Alaska. However, differences are expected in englacial hydrology and subglacial processes because ice is dominantly cold in Greenlandic glaciers. Data currently available from tidewater glaciers in Greenland are insufficient to discuss the aforementioned details of short-term ice speed variations and the

differences compared to other regions. In-situ data near the calving front is sparse owing to the difficulty in installing and operating instruments for long time periods. Accordingly, reported data covers the relatively short summer periods from a single year, or at best a couple of years. Furthermore, previously studied large glaciers are characterized by exceptionally fast ice flow, which makes it difficult to analyze relatively small perturbations superimposed on a large magnitude of background speed.

We present continuous GPS measurements performed within 4 km from the front of Bowdoin Glacier, a tidewater glacier in northwestern Greenland. The measurements were collected for a total of 90 days, spanning six summer seasons from 2013–2019. High frequency positioning data obtained at three locations were processed for short-term variations in horizontal ice speed and vertical displacement. The data revealed details of the glacier dynamics modulated by tide, melt- and rainwater input to the base of the glacier. Mechanisms driving the speed change are discussed based on a frequency analysis and the

correlations of the ice speed with air temperature, tidal height, and glacier surface uplift.

## 2 Study site

Bowdoin Glacier, also known as Kangerluarsuup Sermia in Greenlandic (Bjørk et al., 2015), is one of the marine-terminating outlet glaciers along the coast of Prudhoe Land, northwestern Greenland (Fig. 1a). The glacier is situated at 77°41' N, 68°35' W about 30 km northeast of Qaanaaq Village, feeding ice into Bowdoin Fjord through a 3-km-wide calving front at a rate of

~400 m a$^{-1}$ (Fig. 1b). According to ground-based ice-radar surveys in 2013 and 2014, ice near the front was 264–284 m thick and grounded in water 230–253 m deep, indicating the glacier was very close to flotation (Sugiyama et al., 2015). Satellite-based observations have shown that rapid retreat of Bowdoin Glacier began in 2008, after a greater than two-fold increase in the flow speed occurred between 1999–2006 (Sakakibara and Sugiyama, 2018). The glacier sections located below 850 m a.s.l. have thinned at a rate of 1.11 m a$^{-1}$ from 2001 to 2018, which agrees with a general trend observed in other glaciers in the

region (Wang et al., 2021).

Bowdoin glacier has been the focus of field studies on its ice dynamics and ice-ocean interactions since 2013. Measurements performed on the glacier include ice speed (Sugiyama et al., 2015; Jouvet et al., 2018), surface elevation (Tsutaki et al., 2016), seismicity (Podolskiy et al., 2016), calving (Jouvet et al., 2017; Minowa et al., 2019; van Dongen et al., 2020; 2021), and



englacial ice temperature (Seguinot et a., 2020). In-situ data obtained near the calving front have revealed complex glacier
dynamics and demonstrated its importance in the recently observed rapid glacier thinning and retreat. Measurements were also
performed in the fjord to investigate the upwelling of subglacial discharge (Podolskiy et al., 2021a), acoustic (Podolskiy et al.,
2022) and seismic signals induced by calving (Podolskiy et al., 2021b; 2021c), as well as the impact of glacial meltwater
discharge on geochemical (Kanna et al., 2018; 2020) and biological fjord environments (Matsuno et al., 2020; Nishizawa et
al., 2019). The research results have confirmed a strong link between the glacier dynamics and the ocean, as well as the vital
role of glacial discharge in marine ecosystems.

For several weeks in each summer from 2013–2017 as well as 2019, we carried out field observations on the lower reaches of
Bowdoin Glacier. In addition to the various measurements reported in the previous studies listed above, a high frequency GPS
survey was performed to investigate the glacier dynamics. Following up on a report on the results obtained in 2013 (Sugiyama
et al., 2015), this paper describes the details of the short-term ice speed variations revealed by the data over these six summer
seasons.

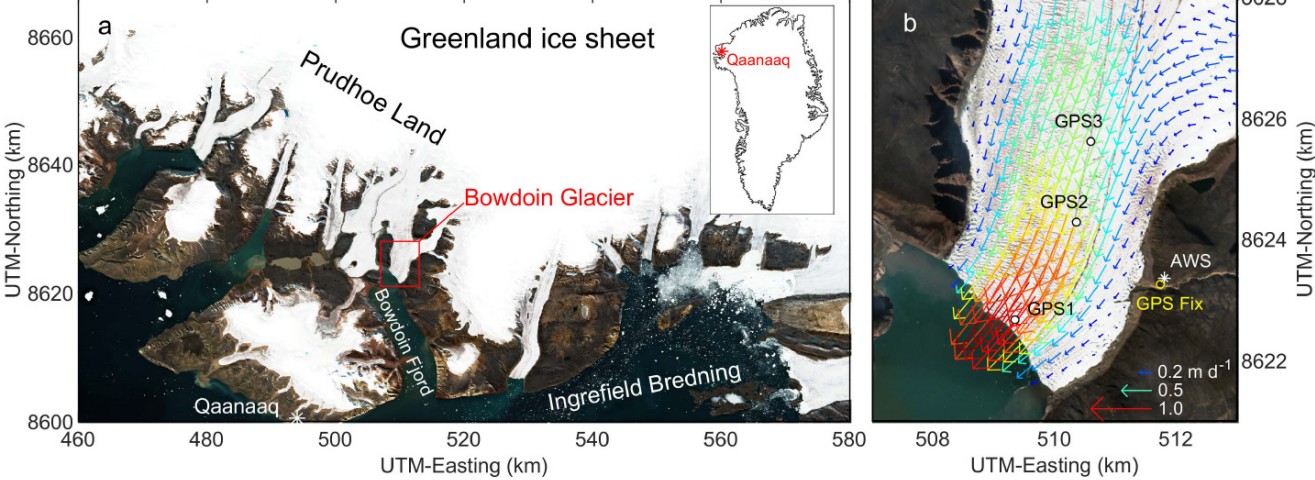

Figure 1: (a) Satellite image (Sentinel 2, July 17, 2019) showing the Qaanaaq region in northwestern Greenland. The red box
indicates the area shown in (b). The inset shows the location of the study site in Greenland. (b) The region of study at Bowdoin
Glacier. The color-coded arrows are surface flow vectors averaged for June 29–August 4, 2019 from the PROMICE Ice Velocity
product (Solgaard et al., 2021). The markers indicate the locations of the survey sites (GPS1–3), GPS reference station (GPS Fix),
and AWS.

## 3 Methods

### 3.1 Ice speed

The GPS ice speed measurements were performed from 5–26 July in 2013, 10–23 July in 2014, 7–19 July in 2015, 5–20 July
in 2016, 7–16 July in 2017, and 2–22 July in 2019. The data acquisition period spanned 9–21 days each year and covered 90
days in total. We installed three GPS receivers (Enabler Inc., GEM-1) on the glacier ~0.5, 2.5 and 4 km from the calving front,
approximately at the same positions every year (Fig. 1b). The receivers were kept running continuously with batteries and





solar panels to record L1 and L2 GPS signals every second. GPS antennae were mounted on the top of 2-m long aluminum poles drilled and frozen in the ice. The GPS data collected with 1-s intervals was post-processed with those from a reference

station installed on the eastern flank of the glacier (Fig. 1b). The processing was performed with the RTKLIB software (https://www.rtklib.com/rtklib.htm) in a static mode to obtain three dimensional coordinates for every 15-minute time window. With baseline lengths shorter than 3 km, the uncertainty of the static GPS positioning is generally several millimeters in the horizontal direction. This uncertainty corresponds to ~10–30% of the ice motion in the 15-minute time window. The uncertainty in the vertical direction was about 5 mm, according to previous measurements under similar conditions (Sugiyama

et al., 2010).

Horizontal ice speed was computed by filtering a time series of the positioning data using a local linear regression routine with a half-width time window of 1 h. To investigate the deviation of the ice motion from a general trend, the mean ice motion was subtracted from the positioning data. The mean ice motion was computed by linear regression of the positioning data obtained in each season. The residual speed and vertical displacement were used to discuss ice speed variations and surface uplift.

Ice velocity distributions over the region of study were obtained for the period spanning 29 June to 4 August 2019 from the PROMICE (Programme for Monitoring of the Greenland Ice Sheet) Ice Velocity product based on Sentinel-1 SAR offset tracking (Solgaard et al., 2021).

## 3.2 Air temperature, precipitation, and tides

During the field campaigns, an automatic weather station (AWS) (Vaisala WXT510 or WXT520) operated on the eastern flank

of the glacier at 100 m a.s.l. (Fig. 1b). The sensor was installed 1.5 m above the ground and connected to a data logger (Campbell Scientific, CR1000) recording air temperature and liquid precipitation every 5 min. The time series of the temperature data were filtered using the same method and time window applied to the GPS positioning data. Hourly cumulative precipitation was used for plotting the data.

Tidal data in the region of study was available at Pituffik/Thule (76°33' N, 68°52' W). Sea surface level is measured by the

Technical University of Denmark and hourly data provided as a part of the Global Sea Level Observing System network (http://www.gloss-sealevel.org). Tides were also measured with a pressure sensor installed near the glacier front for 20 days during the field campaign in 2016 (Minowa et al., 2019). Data comparison shows that the tidal signal at Bowdoin Glacier is in phase with that at Pituffik/Thule, but its amplitude was larger by a factor 1.18 (with a coefficient of determination $r^2 = 0.98$).

## 3.3 Frequency analysis

Frequency analysis was performed on the ice speed, tides, and air temperature data to find frequency components in the temporal variations. A time series of data obtained in each year was converted to the frequency domain by using a fast Fourier transform algorithm implement in the software MATLAB as a function "fft" (Frigo and Johnson, 1998). The sampling intervals of the speed, tide, and temperature measurements were 15 min, 5 min, and 1 hr, which give Nyquist frequencies of 48, 144,



and 12 d$^{-1}$, respectively. The resolution of the frequency analysis is dependent on the length of the sampling period in each
year, which ranged from ~0.05 to 0.1 d$^{-1}$ for our data from the six summer seasons.

In general, ice speed and temperature power-frequency diagrams obtained for each year were not precise enough to discuss
peak frequencies and amplitudes. This was because our measurement period in each year (9–21 days) was not sufficiently long,
given the uncertainty in the GPS coordinates as well as the ambiguity in the periodic variations in the speed and temperature.
Nevertheless, the preciseness of the frequency analysis was improved by stacking spectrums computed for the six summer
seasons and filtering by the local linear regression (Fig. S1). The time windows of the filtering were empirically determined
to be 0.03 d$^{-1}$ for ice speed and air temperature, and 0.01 d$^{-1}$ for the tide, so that high frequency oscillations and noise were
eliminated by the filtering.

### 3.4 Diurnal pattern analysis

Diurnal patterns in ice speed and temperature variations were analyzed by stacking data. First, diurnal signals were extracted
by subtracting daily trends obtained by filtering the original data (Figs S2a–l). Second, a one-diurnal-cycle composite was
constructed for each year by stacking and averaging daily variations (thin curves in Fig. S2m). Finally, diurnal patterns obtained
for the six field campaigns were combined by filtering hourly mean speed and temperature to obtain representative diurnal
variations (thick curves in Fig. S2m).

### 4 Results

### 4.1 Ice speed

Ice speed at the three GPS sites showed temporal variations on timescales from hours to days (Fig. 2). In 2013, the speed at
GPS1 varied within a range from 1.21 to 2.27 m d$^{-1}$, which corresponds to 80–150% of the mean speed over the 16-day survey
period (1.49 m d$^{-1}$) (Fig. 2a). Speed obtained at GPS2 and GPS3 showed similar variations within 80–155% and 75–160% of
the mean values, respectively. Similarly large speed variations were observed in the following years as well. The range of the
speed change at GPS1 was 75–140%, 80–130%, 75–170%, 80–125%, and 70–185% for the measurements in 2014–2017 and
2019, respectively (Figs 2b–f). In general, the ice speed variations over days were associated with change in air temperature.
For example, the glacier decelerated during the first half of the survey periods in 2015 and 2017 in association with a drop in
temperature (Figs 2c and e). Ice speed increased in the second half of the periods corresponding to rising trends in temperature.
In each year, the glacier significantly accelerated once or twice, as consistently observed at GPS1–3. In 2016, for example, ice
speed progressively increased for the first three days and reached a peak speed on 9 July (Fig. 2d). The peak speed at GPS1
(2.83 m a$^{-1}$) was 70% greater than the mean over the measurement period in 2016. Speed rapidly dropped within a day, which
was followed by gradual deceleration for the next ~10 days and smaller acceleration from 19 to 21 July. Similar speed
variations were obtained at GPS2–3. The magnitude of the speed decreased upglacier, but similarly large acceleration (80% at
GPS2 and 90% at GPS3) was observed on 9 July 2016. The peak speed at the three locations occurred simultaneously within





0.5 hr. Short-term speed-up events were observed in 2013 as well. The glacier accelerated by 50–60% on 13 July 2013, which was followed by a less pronounced acceleration that peaked on 18 July (Fig. 2a). The first event coincided with heavy rain with peak precipitation of 5.6 mm hr$^{-1}$ at 19:00–20:00 (hereafter we refer to local time UTC−2), whereas no precipitation but a rapid temperature rise was observed during the second event. Fast ice flow accompanied by rain was also observed on 19 and 21 July 2014 (Fig. 2b).

Superimposed on the trends and events are diurnal and semidiurnal ice speed variations. Semidiurnal variations are evident at GPS1 in 2019 (Fig. 2f). Speed peaked twice a day, approximately in coincidence with or slightly preceding the lowest tides, which we will discuss later. Although the signals were weaker in some years, semidiurnal speed variations were observed every year at GPS1. The semidiurnal signals diminished upglacier towards GPS2 and GPS3. Signals were still observable at GPS2 during the survey in 2013, 2014, and 2019 (Figs 2a, b, and f), whereas they are difficult to find at GPS3.

In addition to the short-term variations described above, it should be noted that the mean speed over each survey period showed year-to-year variations. At the site GPS1, ice flowed relatively faster in 2014 with a mean speed of 1.74 m d$^{-1}$, whereas the mean speed in 2019 was 25% slower (1.30 m a$^{-1}$). The speed at the other two sites showed the same trend, i.e. flows faster in 2014 and slower in 2019. The ranges of the year-to-year variations were similar at GPS1–3. Mean speed in each year varied within 85–115% of the grand mean over the entire six summer seasons (1.52, 1.13, and 0.91 m d$^{-1}$ for GPS1–3).

**4.2 Vertical ice motion**

Vertical coordinates showed generally downward ice motion with mean rates of −13, −38, −32 mm d$^{-1}$ at GPS1–3, respectively (Figs 2g–l). The general trends in the vertical motion at GPS2 and GPS3 were similar every year, i.e. the mean rate in each year falls within ±15% of the total mean over the six summers. However, the mean rate at GPS1 showed significant year-to-year variations represented by a relatively gradual motion in 2016 (−2 mm d$^{-1}$) (Fig. 2j) and the steepest motion in 2017 (−32

mm d$^{-1}$) (Fig. 2l).

Detrended vertical coordinates showed deviation from the general trends typically by several centimeters (Figs 2g–l). Importantly, the detrended vertical displacements obtained at GPS2 and GPS3 were associated with horizontal ice speed. For example, relatively upward displacement from 19 to 21 July 2014 coincided with fast ice flow (Figs 2b and h). Speed-up events in 2013, 2016, and 2019 corresponded to uplift of the glacier surface (Figs 2a and g, d and j, f and l). Association between ice

speed and vertical motion at GPS2–3 was also observed in the general trends in 2015 and 2017 (Figs 2c and i, e and k).

At the lowermost site GPS1, detrended vertical motion was not consistent with those at GPS2–3, and therefore not well correlated to ice speed variations. Surface uplift was not observed at GPS1 during the speed ups in 2014 and 2016 (Figs 2h and j). Disagreement between GPS1 and GPS2–3 is also evident in the data obtained in 2015 and 2017 (Figs 2i and k). Semidiurnal signals were a distinctive feature at GPS1, clearly observed in 2017 and 2019 (Figs j and l). The phase of the

vertical displacement was consistent with ocean tides, suggesting the ice was partially afloat at GPS1. Semidiurnal vertical motion was ambiguous until 2016 and clearly observable since 2017.





**Figure 2. GPS and meteorological data obtained in July 2013–2017 and 2019. (a–f) Ice speed at GPS1–3 (red, blue, and green), tidal height (grey), air temperature (black) and precipitation (bar). (g–l) Vertical displacement (top) and detrended vertical coordinate (bottom) at GPS1–3 (red, blue and green).**




## 5 Discussion

### 5.1 Diurnal and semidiurnal speed variations

Frequency analysis revealed periodicities in the ice speed variations and their association with tides and air temperature (Fig.
3). In agreement with visual inspection, the power spectrum from GPS1 shows clear peaks at diurnal and semidiurnal frequencies (Fig. 3b). The peak frequency of the diurnal signal (1.00 d⁻¹) coincides with peaks in tides (K1 and P1 tides) and air temperature, whereas that of the semidiurnal signal (1.93 d⁻¹) agrees with a major peak in the tides (M2 tide) (Fig. 3a). The power is ~20% smaller, but diurnal signals are also evident in GPS2 and GPS3. In contrast to the diurnal speed variations persistently observed at the three locations, semidiurnal variations are less significant in the upper reaches. The spectrum from
GPS2 shows a peak at nearly the same frequency (1.94 d⁻¹), but the power is ~70% smaller than that obtained for GPS1. The power of the semidiurnal signal further decreases at GPS3.

The frequency analysis shows a tidal influence on the ice flow speed of Bowdoin Glacier. The magnitude of the speed change diminishes upglacier, consistent with previous studies at Greenlandic tidewater glaciers. Tidal flow modulation decayed exponentially upglacier and no evidence of its influence was found at >10 km from the calving front of Jakobshavn Isbræ
(Podrasky et al., 2012), whereas the influence extended more than 12 km at Helheim Glacier (de Juan et al, 2010). Presumably, the extent of the tidal influence is dependent on geometric and boundary conditions, which are relevant to the force budget near the glacier front, e.g. glacier width, ice thickness, and basal drag. In the case of the 3-km wide Bowdoin Glacier, tidal influence was still observable 4 km from the ocean.

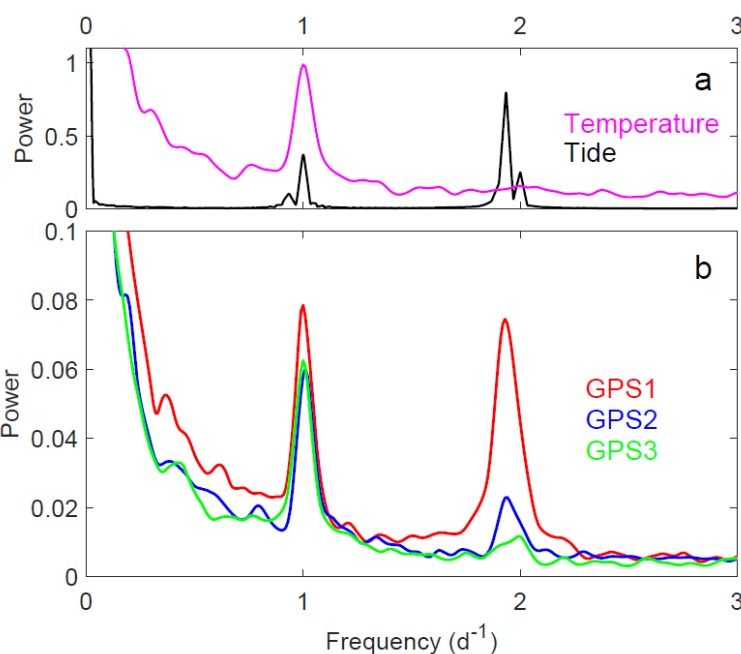

**Figure 3. Power spectral diagrams for (a) air temperature (magenta), tide (black), and (b) ice speed at GPS1–3 (red, blue, and green) obtained by stacking and filtering spectrums for data in 2013–2017 and 2019 (Fig S1).**





An important finding here is that diurnal variations were consistently observed at the three GPS sites, whereas the tidally induced semidiurnal variations diminished from GPS1 to GPS3. This result strongly suggests that a large portion of the

observed diurnal speed variations are not due to tide. We assume that the main driver of the diurnal variations was meltwater production and its influence on the basal sliding. A link between melt and flow speed was assessed by comparing diurnal patterns in air temperature and ice speed at GPS3. Speed from GPS3 was used for this purpose because tidal modulation is small at this distance from the calving front (Fig. 3b). The diurnal ice speed peak obtained by stacking all available data lagged behind the temperature peak approximately by two hours (Fig. 4). We attribute this lag to the time required for the transfer of

surface meltwater to the glacier bed. Over the region extending 37 km from the calving front of Helheim Glacier, diurnal speed peaks lagged behind maximum daily insolation by 6.5 hr (Stevens et al., 2021). Taking the lag between air temperature and insolation maxima into account, Bowdoin Glacier responded to meltwater production more swiftly than Helheim. A likely reason for the relatively short response time at Bowdoin is the smaller dimensions of the glacier. Ice thickness at our measurement sites was 270–330 m (Sugiyama et al., 2015), whereas ice is >600 m thick near the front of Helheim Glacier.

Moreover, the ablation area of Helheim Glacier extends several tens of kilometers from the glacier front. Presumably, meltwater on Bowdoin Glacier drained to the bed more quickly and immediately affected the glacier dynamics in the area of study.

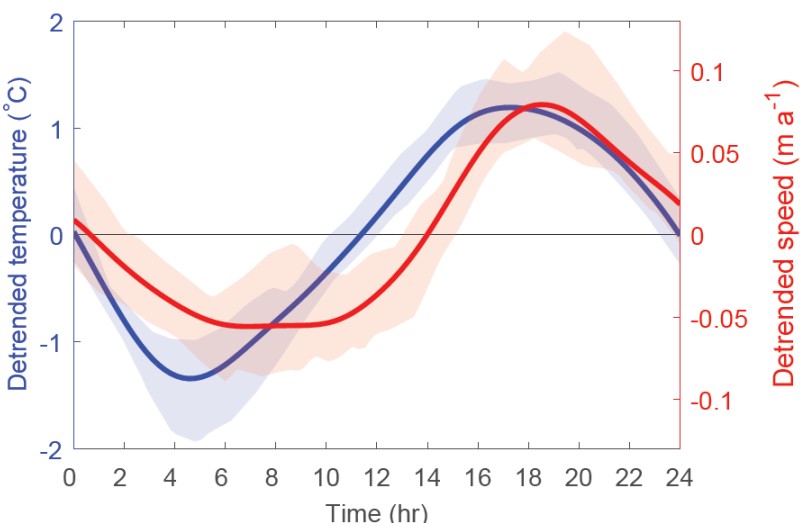

**Figure 4. Diurnal variations in temperature (blue) and ice speed at GPS3 (red) obtained by stacking and smoothing detrended data in 2013–2017 and 2019 (Fig S2). The shades indicate the ranges of year-to-year variations in the diurnal curves.**



## 5.2 Meltwater induced speed variations

The link between meltwater production and ice speed was further investigated by comparing detrended ice speeds with air temperature as a proxy of surface melt rate (Fig. 5). As suggested by the time series (Figs 5a–f), ice speed variations correlate with air temperature (Figs 5g–i). The relationships are slightly different in each year and at each site (Fig. S3), but in general the glacier accelerated as temperature increased. The ice speed is not a linear function of temperature, i.e. glacier acceleration is more pronounced when temperature exceeded ~10°C.

The correlations between ice speed and temperature demonstrate the strong influence of meltwater production on the dynamics of a Greenlandic outlet glacier. Surface meltwater swiftly drains to the glacier bed and affects basal sliding by elevating subglacial water pressure. The immediate impact of melting on the ice speed is evidenced by the relatively short two-hour lag between the temperature and speed peaks (Fig. 4). The importance of water pressure in sliding speed has been demonstrated by borehole measurements in alpine glaciers (e.g. Iken and Bindschadler 1986; Jansson, 1995; Sugiyama et al., 2003) and a similar basal process has been suggested for Greenlandic glaciers based on surface measurements (Podrasky et al., 2012; Davis et al., 2014; Sugiyama et al., 2015; Stevens et al., 2021). Glacier dynamics are sensitive to meltwater input, particularly near the front of a calving glacier where effective pressure is low under the influence of proglacial water (Kamb et al, 1994; Sugiyama et al., 2011). Presumably, the nonlinear response of the ice motion to temperature arose because of such a condition under the region of study.

Since tidal influence is greater near the calving front, the temperature dependence of the ice speed is more ambiguous at GPS1 (Fig. 5g). Moreover, episodic speed-up events do not follow the general relationship with temperature. For example, data points deviate from the trend (Figs 5g–i) during the speed up events in 2013 and 2016 (Figs 5a and d). Further, the speed-temperature relationship varies in each year (Figs 5g–i). These observations indicate that glacier motion is not a simple function of surface meltwater production (e.g. Bartholomaus et al., 2008). Amount and timing of meltwater transfer to the glacier base are affected by glacier surface and englacial drainage conditions. Moreover, the impact of subglacial meltwater input to the basal water pressure is highly dependent on the subglacial drainage efficiency, which evolves in time over the course of the melt season.



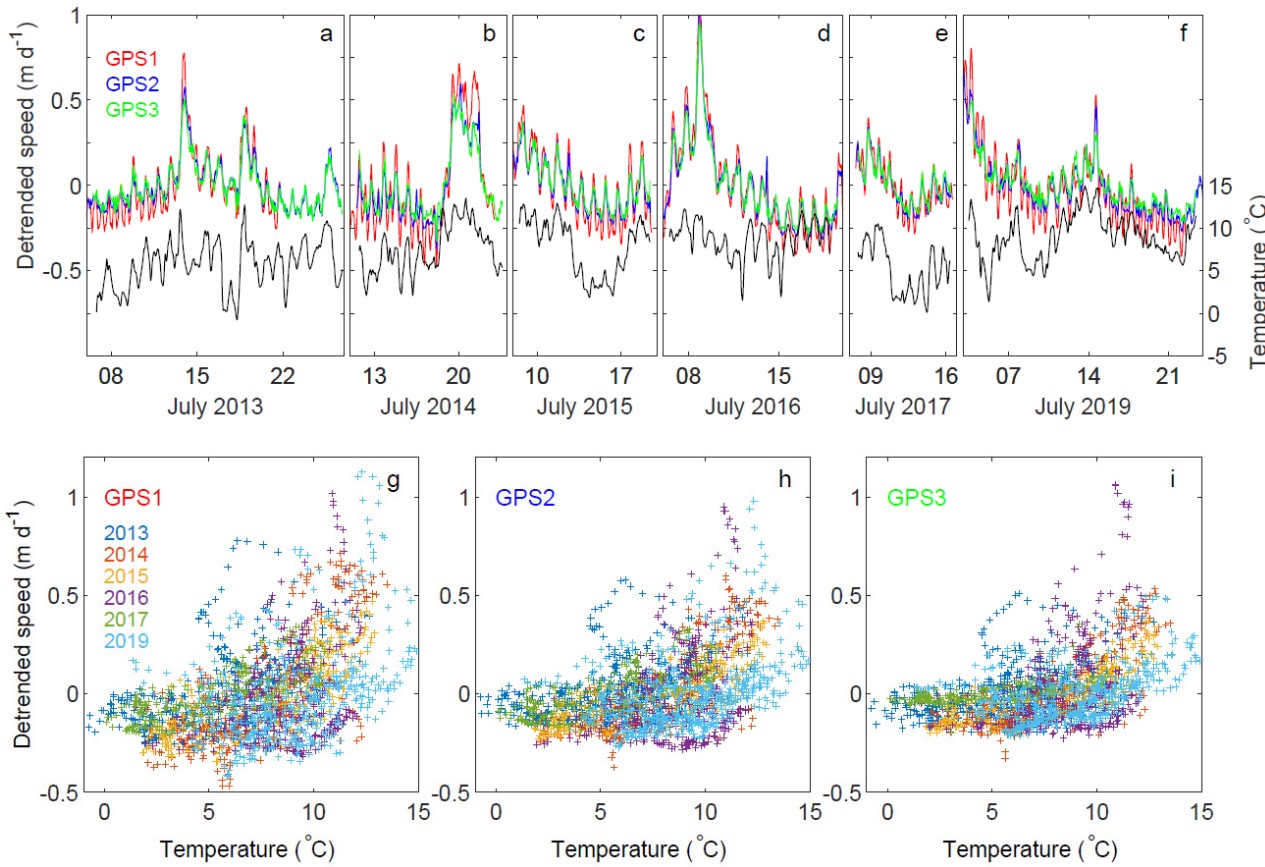

**Figure 5. (a–f)** Detrended ice speeds at GPS1–3 (red, blue, and green) and air temperature. **(g–i)** Scatter plots of detrended hourly mean ice speed at GPS1–3 and temperature. Marker colors indicate the year of the measurements.

## 5.3 Speed up due to rain

It was not only meltwater which affected the subglacial water pressure and sliding. Rain events induced significant glacier accelerations as well. Rain induced glacier acceleration was documented in alpine glaciers (Gudmundsson et al., 2000; Mair et al., 2001; O'Neel et al., 2001; Horgan et al., 2015), and reported from land-terminating sectors of the Greenland ice sheet (Doyle et al., 2015). Nevertheless, detailed observations are sparse for tidewater glaciers in Greenland. Considering the increasing trend for liquid precipitation in Greenland (Niwano et al., 2021; Box et al., 2023), speed up due to rain is of critical importance for ice discharge from the ice sheet.

The glacier responded differently to precipitation events in 2013, 2014, and 2017 (Fig. 6), which implies the complexity of glacier dynamics under the influence of water input, subglacial drainage efficiency, and tides. In 2013, intensive precipitation on 13 July at 19:00–20:00 resulted in abrupt acceleration at GPS1–3 with peak speeds several hours later at 23:00–24:00 (Fig.



6a). The speed up was clearly induced by rainwater, but we suggest that meltwater and tide also played roles. At 15:00, four hours before the rain event, air temperature showed a relatively high daily peak temperature (12°C), suggesting substantial melting on the glacier. Furthermore, the peak speed coincided with low tide, which facilitates fast glacier flow as discussed in the next subsection. Therefore, we assume that a large amount of meltwater and falling sea level enhanced the response of the glacier speed to the rain event in 2013.

In 2014, a substantial amount of rainfall was recorded on 19 and 21 July (Fig. 6b). Ice speed increased on 19 July in coincidence with the first rain event and a fast-flowing condition was kept for three days until 21 July. The mean speed from 19 to 21 July (2.18 m d⁻¹) was ~40% greater than the mean before the acceleration. The consistently high flow speed can be ascribed to relatively warm atmospheric conditions (generally above 10°C from 19 to 21 July), but our data suggests the rain events triggered the speed up and maintained the fast flow until the glacier rapidly decelerated on 22 July. It is interesting that the glacier was largely accelerated by the relatively small amount of precipitation on 19 July (4.8 mm), when a 4x greater amount of precipitation on 21 July (18.9 mm) did not promote further acceleration. Most likely, the subglacial water pressure was relatively insensitive to the second rain event because the basal drainage system developed and became more efficient after the first event.

Precipitation of 10.2 mm on 10 July 2017 caused no significant change in speed (Fig. 6c). During this period, the glacier progressively decelerated with diurnal and semidiurnal signals, which were in phase with ocean tides. The glacier was insensitive to the rain likely because the subglacial drainage system had already developed by this time in the season. We assume that rainwater swiftly drained through the system without significant impact on subglacial water pressure.

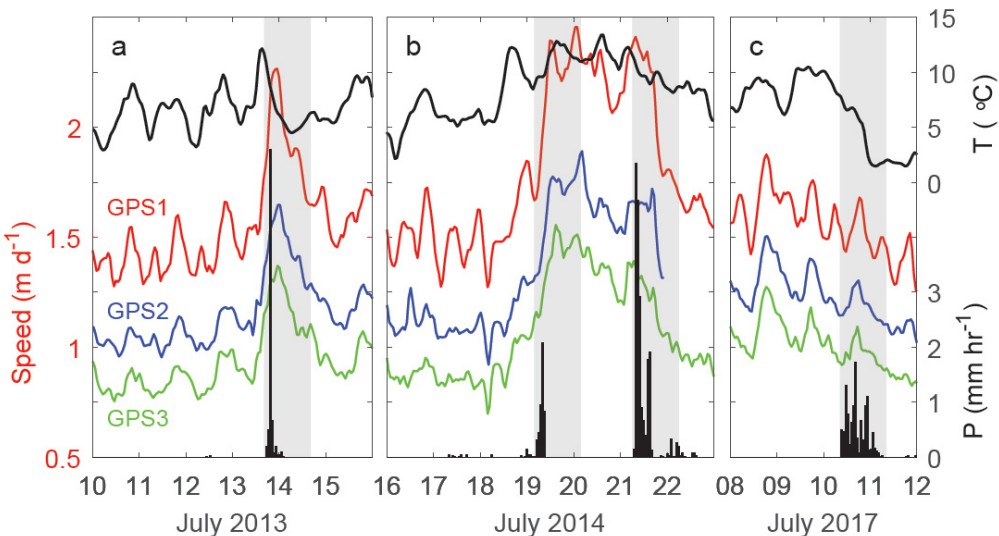

**Figure 6. Ice speed at GPS1–3 (red, blue, and green), air temperature (black line) and precipitation (bar graph) in (a) 2013, (b) 2014, and (c) 2017. Rain events are highlighted by the gray shaded portion.**





### 5.4 Tidal modulation

To investigate the details of the tidal influence on glacier dynamics, ice speed at GPS1 is compared with tidal height during periods of relatively stable temperature and no precipitation (5–9 July 2013, 10–18 July 2014, and 15–22 July 2019) (Fig. 7).

Ice speed approximately anticorrelated with tides, i.e. peak speeds occurred at around low tides (Figs 7a–c). The relationship between speed and tide (Figs 7d–f) are characterized by; (i) peak speeds at or slightly before low tides, (ii) loop trajectories that arise from faster/slower flow during falling/rising tides, and (iii) two modes in the loop trajectories that consist of relatively slow speed changing over the full tidal rage (cyan in Figs 7d–f) and fast speed over the lower tidal range (magenta in Figs 7d–f).

Fast glacier motion during low tides implies that hydrostatic pressure acting on the glacier front is the dominant influence. Reduced ocean water pressure is transferred to the glacier base and side margins, which enhances downglacier ice motion due to basal sliding and ice deformation. The anticorrelation of speed and tides is consistent with observations in Alaska of Columbia Glacier (Walters and Dunlop, 1987) and Le Conte Glacier (O'Neel et al., 2001). Tidal influence on the subglacial water pressure is likely insignificant because such an influence implies fast glacier motion during high tide. The data from

Bowdoin Glacier shows that some of the speed peaks preceded the lowest tides. Observations in other tidewater glaciers in Greenland showed a similar tendency, i.e. peak speeds occurred during falling tides several hours before the minima in tidal height (de Juan et al., 2010; Walter et al., 2012; Podrasky et al., 2014). We attribute this shift to elastic response of the glacier as explained below.

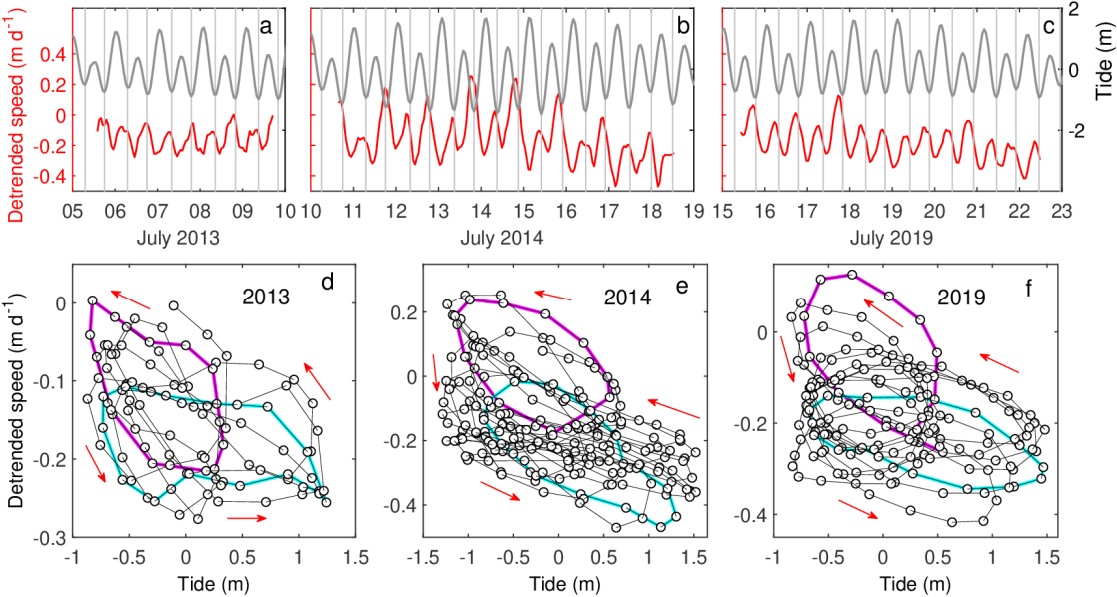

**Figure 7. (a–c) Detrended ice speed at GPS1 (red) and tidal height (black). (d–f) Scatter plots of detrended speed and tidal height. The arrows in d–f show the time sequence of the data. Curves highlighted in magenta and cyan represent the two different loop trajectory groups described in the text.**





If we consider ice deformation over a timescale of a few hours to a day, because of its viscoelastic nature, a substantial portion of the deformation would be due to elasticity (e.g. Sugiyama et al., 2007; Christmann et al., 2021). The rate of elastic

deformation is related to the changing rate of stress. Therefore, contribution of the elastic deformation to ice speed is the greatest when the rate of falling tide is maximum, whereas that of the viscous deformation is greatest at the lowest tide. Considering the 90° phase difference (~3 hr for semidiurnal tidal cycles) between the viscous and elastic deformation, viscoelastic glacier response to the changing hydrostatic pressure explains the loop in the speed-tide trajectory and slight shift in the peak speed.

The bimodal loops in the speed-tide trajectory are generated by the sequence of major and minor peaks in ice speed and tide. During our measurement periods, tidal peaks around midnight were higher than those around noon (Figs 7a–c). Because ice speed peaks in the afternoon are generally greater than those in the morning, major speed peaks occur after minor tidal peaks. Accordingly, speed-tide trajectories associated with major tidal peaks appear in the lower half of the diagrams (cyan in Figs 7d–f), whereas those with minor peaks appear in the upper left of the diagram (magenta in Figs 7d–f). The complexity arises

since ice speed was affected not only by tide but also by meltwater. The melt-induced diurnal variations shown in Figure 4 are superimposed on the tide-induced semidiurnal variations, thus semidiurnal speed peaks in the afternoon are greater than those in the morning.

To illustrate this interpretation, we show that the relatively complex ice speed variations near the front of Bowdoin Glacier can be reproduced by a simple model, which takes into account viscous and elastic responses of ice deformation to tidal

variations and basal sliding modulated by meltwater input. Observed diurnal variations in tide $h(t)$ and air temperature $T(t)$ are approximated as:

$$h(t) = \cos(4\pi t) + \frac{1}{2}\cos(2\pi t) \quad \text{and} \tag{1}$$

$$T(t) = \cos\{2\pi (t + \Delta t_1)\} \quad , \tag{2}$$

where $\Delta t_1$ is introduced for the phase shift of the temperature variation (Fig. 8a). We assume the ice speed perturbation $\Delta u$ consists of three components; $\Delta u_v$ the viscous response of ice deformation given by a linear function of negative tidal height, $\Delta u_e$ the elastic response of ice deformation linearly correlated with the negative tidal rate, and $\Delta u_s$ the basal sliding given by a linear function of temperature.

$$\Delta u (t) = \Delta u_v + \Delta u_e + \Delta u_s = a_1(-h) + a_2\left(-\frac{\partial h}{\partial t}\right) + a_3 T(t + \Delta t_2) \tag{3}$$

The coefficients $a_1$, $a_2$, and $a_3$ are adjusted so that the amplitude of the sliding perturbation is 50% of those of the viscous and elastic perturbations (Fig. 8b). Based on observation (Fig. 4), the phase shifts $\Delta t_1$ and $\Delta t_2$ are chosen so that peak temperature and sliding occurs at 16:00 and 18:00, respectively. Despite the simplicity of the model, as well as minimal adjustment of the parameters, the essential features of the speed-tide trajectory are reproduced by the model (Fig. 8c as compared to Figs 7d–f). The more complex and variable relationship between speed and tide can be attributed to day-to-day variations in the phase



shifts and amplitudes of the oscillations in Equations (1)–(3). The non-linear relationship between temperature and speed (Figs
        5g–i) should be also considered for more accurate modeling.

        The timing of the tidal modulation at Bowdoin Glacier is similar to that observed in Alaska and other glaciers in Greenland,
        but considerably different from those reported in Antarctica. In the case of outlet glaciers in Antarctica, peak speeds typically
        coincide with a falling tide rather than the lowest tide (Anandakrishnan et al., 2004; Gudmundsson, 2006; Murray et al., 2006;

Marsh et al., 2013). Presumably, the elastic dynamics of ice shelves play a dominant role in the short-term ice speed variations
        in Antarctica, whereas viscous behavior is more important at grounded glaciers. Further, stick-slip ice motion and subglacial
        sediment deformation add additional complexity to the dynamic response of Antarctic glaciers to tides (Bindschadler et al.,
        2003; Walker et al., 2012).

        Based on numerical modeling as well as field observations, Christmann et al. (2021) demonstrated the importance of

viscoelasticity for the dynamics of fast-flowing glaciers in Greenland. They argued that up to 30% of ice deformation occurring
        along the flowline of Nioghalvfjerdsbr is attributable to elastic strain. Our data and the conceptual model provide an insight
        into how to take viscoelastic glacier response into account for accurate modeling of short-term ice speed variations induced by
        tide and meltwater.

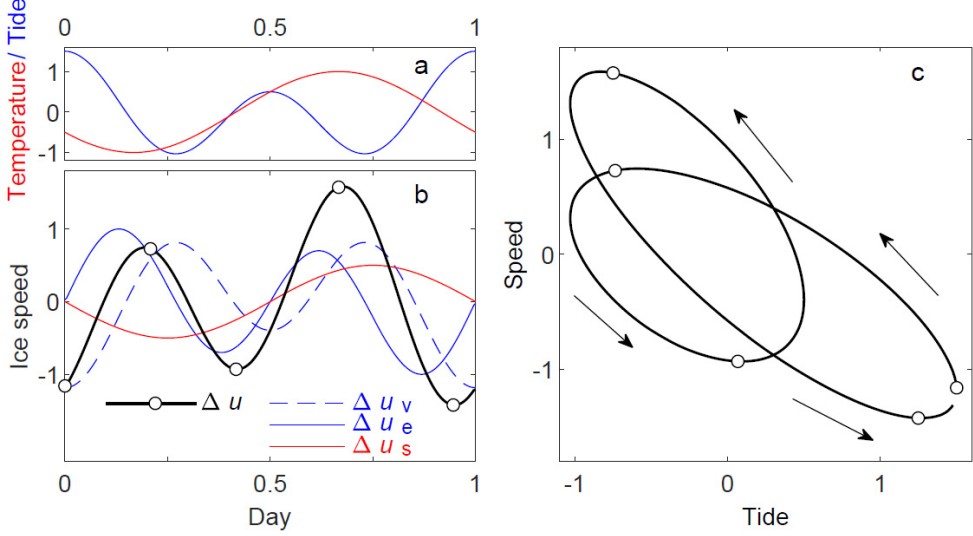

**Figure 8. (a) Diurnal variations in tidal height (blue) and temperature (red) prescribed by Equations (1) and (2). (b) Ice speed
        computed by Equation (3) (black) as the sum of the perturbations due to viscous (dashed blue line), elastic (solid blue line) responses
        to tide, and sliding enhanced by meltwater (red). The open circles show local minima and maxima, which correspond to those
        indicated in (c). (c) Diurnal variations in ice speed and tidal height. The arrows indicate the time sequence of the data. The variations
        in the tide, temperature and ice speed are in arbitrary units.**




## 5.4 Uplift during speed up

The vertical component of glacier motion showed short-term variations associated with ice speed change (Figs 9a–f). The correlation between speeds and detrended vertical coordinates is clearly observed at GPS2 and GPS3 (Figs 9h and i). Ice flows faster when the glacier surface is elevated, although only several centimeters above the general trend. Similar correlations are
consistently observed in the plots for individual years (Figs S4m–x). In contrast to the upper two GPS sites, no clear relationship is observed at the lowermost site GPS1 (Fig. 9g and Figs. S4g–l)

Vertical motion of the glacier surface arises from three processes: sliding over an inclined bed, vertical straining, and basal separation (Hooke et al., 1989). The generally downward trends (Figs 2g–l) can be attributed to the first two processes, i.e. sliding along a downglacier sloping bed and ice thinning due to a longitudinally stretching flow regime. Nevertheless, the
observed uplift during speed up cannot be explained by sliding because excess downward motion is expected when sliding speed increases. Similarly, vertical straining is excluded from the mechanism since the stretching flow regime is enhanced by greater acceleration near the glacier front (Podolskiy et al., 2016). Therefore, our data demonstrates that acceleration of the glacier was associated with basal separation due to formation or enlargement of subglacial water cavities. In previous studies in alpine glaciers, surface uplift has been ascribed to basal cavity formation (e.g. Iken et al., 1983; Kamb and Endgelhardt,
1987; Sugiyama and Gudmundsson, 2003), which facilitates fast sliding by reducing basal traction and by the hydraulic jacking mechanism (Lliboutry, 1968; Iken, 1981; Röthlisberger, 1981). Near the front of a calving glacier, ambient subglacial water pressure is high under the influence of proglacial water (Kamb et al., 1994; Sugiyama et al., 2011). This is a favorable condition for a pressure increase above flotation, uplift due to basal separation, and acceleration of the glacier.

The lack of correlation for GPS1 can be attributed to the ice thickness close to flotation near the glacier front. According to
the ice radar measurements in 2013 and 2014, the calving front of Bowdoin Glacier was grounded, but nearly 90% of the entire ice thickness was below sea level (Sugiyama et al., 2015). Ice near the front was thinning at a rate of ~5 m a$^{-1}$ in 2007–2013 (Tsutaki et al., 2016), suggesting that the glacier front became afloat during our study period. This is consistent with the tidally modulated vertical ice motion observed after 2017 (Figs 9e and f). Given that the ice was afloat, and the glacier base was hydraulically connected to the ocean, surface meltwater input would not cause pressure increase above the flotation. At GPS1,
uplift during speed up was only observed in 2013 (Figs S4a and g), so we assume the hydraulic connection to the ocean has been established since 2014.



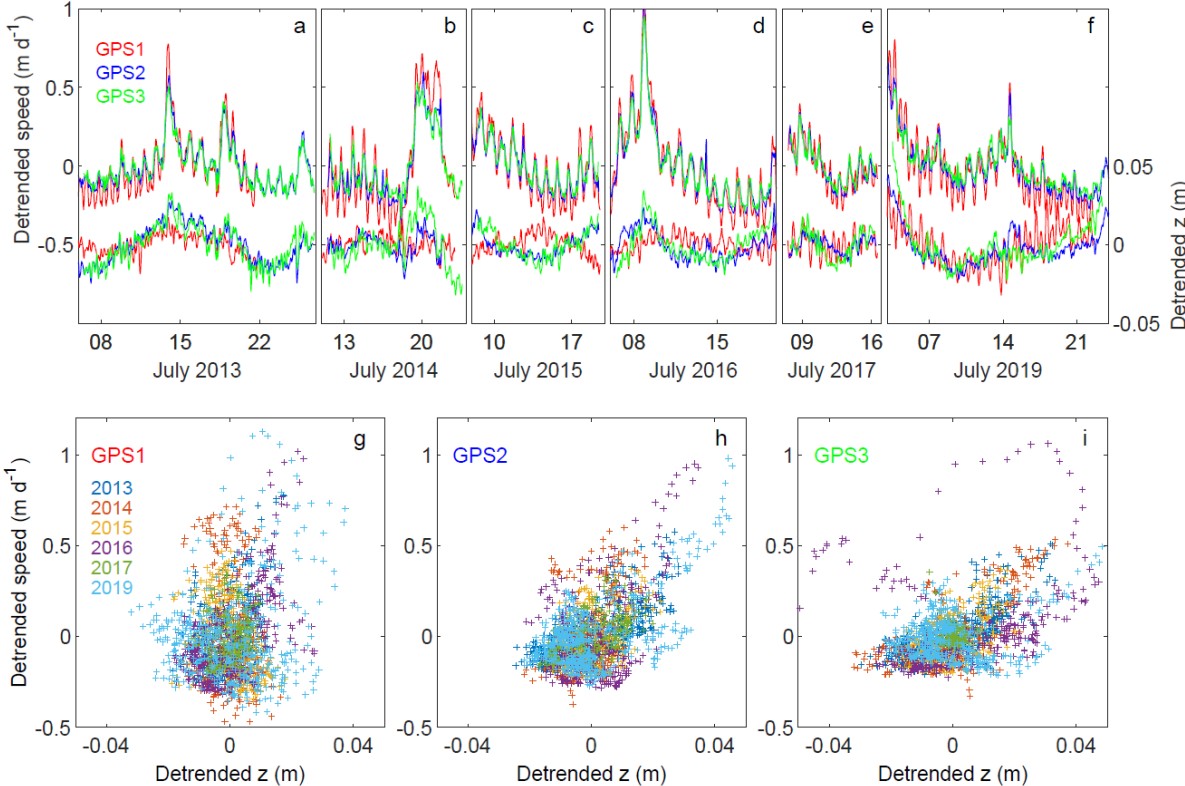

**Figure 9. (a–f) (top) Detrended ice speed and (bottom) vertical coordinate at GPS1–3 (red, blue, and green). (g–i) Scatter plots of detrended hourly mean ice speed and vertical coordinate at GPS1–3. Marker colors indicate the year of the measurements.**


## 6 Conclusions

To investigate short-term ice speed variations, high frequency GPS measurements were performed near the front of Bowdoin Glacier, a tidewater glacier in northwestern Greenland. The measurements were carried out for several weeks in July from 2013–2017 and 2019 at three locations within 4 km from the glacier front. Horizontal ice speed showed temporal fluctuations

with timescales of hours to days, which were associated with ocean tides, air temperature variations, and precipitation events. Frequency analysis indicated diurnal and semidiurnal periodicities in the speed variations. The diurnal signal was consistently observed at all three GPS sites, whereas the semidiurnal signal diminished upglacier. The result implies that diurnal variations driven by meltwater input are superimposed on tidally modulated diurnal and semidiurnal variations. The tidal influence decays within several kilometers from the calving front. The diurnal speed peaks lag approximately two hours behind the daily

temperature maxima, implying a swift transfer of surface meltwater to the glacier base and immediate impact on basal sliding.



In addition to diurnal variations, we found a correlation between ice speed and air temperature variations of the timescale of days. The glacier significantly accelerated during a period of temperatures higher than 10°C. Additionally, speed-up events were observed after precipitation events. These observations confirm a strong link between surface melt- and rainwater and subglacial water pressure. The temperature dependence of ice speed variations was different each year because the timing and magnitude of the meltwater influence on the glacier dynamics is affected by the evolution of englacial and subglacial drainage systems.

Semidiurnal ice speed peaks coincide with or slightly precede low tides. Ice flows faster during falling tides, resulting in loops in the speed-tide trajectory. Based on these observations, we assume that the tidal modulation was generated by the viscoelastic ice mechanics responding to changing hydrostatic pressure acting on the calving front. The observed diurnal and semidiurnal speed variations were reproduced by a simple linear model, which considers viscous and elastic ice deformation in response to tidal height and basal sliding enhanced by diurnal meltwater input.

With the exception of the lowest site where ice was afloat, the GPS data showed upward ice surface motion during fast flowing periods. The uplift associated with speed up is consistent with previous observations in alpine glaciers, where it has been attributed to basal separation due to pressurized water cavity formation. This result confirms the impacts of melt and rainwater input on the subglacial hydraulic conditions, which in turn facilitate enhanced basal ice motion.

Our high frequency GPS data revealed that the dynamics of a Greenlandic tidewater glacier is strongly affected by surface water during the melt season. Evolution of englacial and subglacial drainage systems give rise to additional control of melt- and rainwater-driven ice speed variations, as commonly accepted in alpine glaciers. Despite the similarities with land-terminating mountain glaciers, tidal influence results in more complex short-term variation in the dynamics of tidewater glaciers. Importantly, tide-induced flow variations in Greenland show significant differences from those reported in Antarctica in terms of timing and inland extent. In addition to its direct impact on ice discharge, ice dynamics near the tidewater glacier front is crucial for calving because calving is preconditioned by the formation and opening of crevasses. Timing, frequency, and magnitude of calving are potentially controlled by flow variations due to tide, melt, and rain. Thus, the results of our study provide important insights into tidewater glacier dynamics and contribute to an accurate understanding of future evolution of the ice sheet under a changing climate and environment in Greenland.



## Data availability

The presented data will be available online upon acceptance of the paper.

## Author contribution

S.S. designed the study and coordinated the field study with contribution by G.J. and M.F. S.S., S.T., D.S., I.A. K.K., E.P. G.J. and M.F. performed measurements on Bowdoin Glacier. Y.W. analyzed satellite data. S.S. processed the data, produced the figures, and wrote the manuscript. All authors contributed to discussion and manuscript preparation.

## Competing interests

One of the authors (Evgeny Podolskily) is a member of the editorial board of The Cryosphere.

## Acknowledgements

We thank members of the field campaigns at Bowdoin Glacier from 2013–2017 and 2019 for their contribution in the field. Special thanks to T. Ohshima, K. Petersen and S. Daorana for providing logistic support in Qaanaaq. This research was funded by MEXT (Japanese Ministry of Education, Culture, Sports, Science and Technology) through the Green Network of Excellence (GRENE) Arctic Climate Change Research Project, the Arctic Challenge for Sustainability (ArCS)
(JPMXD1300000000), ArCS II projects (JPMXD1420318865), and JSPS KAKENHI grant number 20H00186 (2020–2025). The presented data will be available online upon acceptance of the paper.

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
