# Peer review of "Ice speed of a Greenlandic tidewater glacier modulated by tide, melt, and rain"

_EGUsphere, 2024_

## Referee Comment (RC2)

Review of '**Ice speed of a Greenlandic tidewater glacier modulated by tide, melt, and rain**'
by Sugiyama et al., for *The Cryosphere*

Summary

This manuscript investigates the short-term drivers of speed variability at Bowdoin Glacier, a medium sized outlet glacier in Greenland, using three deployed GPS receivers during six summers from 2013-2019. 3D ice motion is compared to tidal fluctuations and AWS-recorded near-surface air temperature and precipitation. Units were deployed at approximately the same location each year and oriented along flow with one unit nearest the terminus, and the third ~4 km inland. The study finds that glacier speeds responded both diurnally (melt-driven signal) and semi-diurnally (tide-modulated), though the later forcing decayed in influence with increasing distances from the front. Rain-driven acceleration was also detected in some years, though inconsistently across years, which is attributed to the subglacial drainage system evolution and dependency on the state of the subglacial system when precipitation events occur. Tidal influence was strongest near the front, and most pronounced once the terminus is believed to be at or near flotation heights. Most interestingly, this study shows that the 2 inland GPS receivers record uplift during periods (up to multi-day) of acceleration on the order of several centimeters, which is linked to the physical separation of the glacier from the bed as pressurized subglacial drainage systems form.

The manuscript is well written and arranged in a comprehensive and logical structure, with appropriate figures that complement the main results in the text. The study presents important results for understanding drivers and response times of dynamic outlet glaciers and evolving subglacial systems and offers valuable in situ observations that capture processes that occur at higher frequency than can be captured by most remote sensing studies. This manuscript is therefore nearly suitable for publication in TC in its current form, but I find there to be two topics that warrant further analysis/discussion in the main text, mainly: context on the position and phase of the terminus throughout the study period and (2) more figures that describe how key variables (such as lag time and coefficients of temp/speed relationships) vary between years and any trends that were observed. These themes are discussed below in the 'Main', followed by minor comments and requests for clarification.

Main

*Terminus change*: Some general description of Bowdoin's terminus change are provided in the background, but it would be beneficial to include more information on terminus change during the study period, especially because variations in distance-to-front is found to be an important component on varying responses to tidal impacts between sites 1, 2 and 3. Similarly to the phase of tidal change, the phase of terminus change (whether advancing of retreating), and distance from the nearest GPS receiver (unit 1) each year, and the range in that distance over the seasonal study window, are all important variables that may lend more context to various signals detected during the 2013-2019 period. While I believe the terminus remained relatively stable after 2013 as compared to the large retreat in the preceding years, the interannual variability in when advance/retreat occurs (if any) would still be important to address in this manuscript.

*Interannual changes in key variables*:
A main strength of this manuscript over previously published studies at this glacier is the extended 2013-2019 study period, which enables the authors to investigate trends and interannual variability in key characteristics of ice flow. While the 'stacking' approach across multiple years was necessary to conduct the fast fourier transform and identify the dominant frequency components, multiyear mean values (such as shown in Figure 4) exclude potentially informative information on how seasonality and characteristics of low have evolved over time. For example, the temperature-max speed lag time of 2 hours was only provided for GPS3, and using a mean result from stacked daily values. It would be useful to understand how this lag compared across years at GPS3, or even compared to lag time at GPS2. Another recommendation on this theme would be to provide corresponding text in the main manuscript that describes the relationships seen in the scatter plots (which include all years superimposed). The plots by themselves are not super informative, and difficult to discern how correlations vary (or remain consistent) across years.

Minor
*Request for more clarity*: Were 2013, 2014 and 2017 the only years where precipitation was recorded during the study period?

*Scatter-plots* (for example, in figure 9): Consider using a colormap that avoids very similar colors. It is difficult to discern the years shown in light blue and darker blue (2013 and 2019).

*Line 80*:
Consider replacing "continuous" with "multi-year year series".

*Request for additional citation*: I think some of the introductory discussion on subglacial hydrology is light on citations, particularly for more recent work. Should also include citations for discussion on tidal force balance at the calving front. (lines 35-42, and again supporting citations in lines 72-80). There are some important citations used later in the discussion of results on subglacial hydrology that could be incorporated again in the earlier introduction/background.

*Request for additional clarity*: I found some of the detrending methods description confusing. For example, in some places the text uses "detrended" to refer to, what I believe, is simply the time series with the seasonal mean subtracted. In Figure 5, 2019 speeds certainly show a "trend" over the July period, though the mean is zero. However, I assume this detrending approach (where a seasonal acceleration or deceleration may be present) is a different approach used to the "stacking" described, where a mean diurnal speed is computed. Can you please provide more clarity on these methods?

Line 170 – how is *significant* acceleration defined here? Based on a threshold rate of change?

---

## Author Comment (AC2)

RC1: 'Comment on egusphere-2024-1476', Anonymous Referee #1, 18 Jun 2024

This paper uses detailed GPS-based ice velocity records from near the terminus of Bowdoin Glacier, NW Greenland, to explore controls on the short term variability in ice motion along the lowermost 4 km of the glacier. This methodology is quite similar to that employed in previous studies of the dynamics of alpine glaciers and outlet glaciers from the Greenland Ice Sheet, and correspondingly some of the findings are quite familiar from earlier work. However, studies on the short term dynamics of tidewater glaciers remain comparatively uncommon, and this study thus adds value in demonstrating how far our understanding of glacier dynamics applies (and doesn't apply) in this context. The GPS data on which the paper is based is of high quality and the analysis and interpretation seem largely sound and well supported. I have no major concerns but just a few minor comments:

Thank you very much for reading our manuscript and providing helpful comments. Below (in red), we address the reviewer's comments and explain how we revise the manuscript.

L31. Can you be more quantitative on the relative contribution of these processes?

Yes, we will introduce the relative contributions reported in previous literature.

L33. I think it's overstating things to say that this is 'key' (implying it's the single most important factor) – it is one many topics that are important in understanding the current and future mass loss of the ice sheet.

We agree with this comment. The text will be revised.

L249. More so than simply the time taken for water to get to the bed, I think an important consideration here is what controls the timing of the diurnal peak in water pressure. Throughout the middle part of the day when melting is greatest, meltwater input to the glacier will exceed meltwater discharge from the glacier, causing water storage and pressure to increase (the opposite occurs during the remainder of the day). Thus the period of maximum pressure will not likely coincide with maximum melt rates, but rather will occur slightly later (towards the end of the higher melt period of the day), as you observe (see for example Cowton et al 2016).

We agree with the comment. The text will be revised by including the process controlling

water pressure.

L255-257. Could snow cover also be a factor here? I assume there was little in the study area at Bowdoin Glacier, but if the study area at Helheim extended 37 km from the terminus, then perhaps remaining snowpack could have played a role in slowing the runoff of meltwater.

Meltwater input from snow-covered upper regions is possible both in Helheim and Bowdoin Glaciers, but delay in runoff due to temporal retention in snowpack is expected to be longer than ~hours. It obscures diurnal ice speed variations, but it is unlikely to cause a "shift" in diurnal peaks. Further, Most of the GPS stations on Helheim Glacier was below 800 m a.s.l. (Stevens et al., 2021), which is well below the equilibrium line altitude. Therefore, the studied area was snow free during the mid and late summer survey period (July and August). Because of the reasons above, we excluded the influence of snowpack from the discussion here.

L282. Suggest 'The amount and…'

We correct the text as suggested.

L283. Suggest '…subglacial meltwater input on the basal water pressure…'

We correct the text as suggested.

L283. It's not totally clear whether Bartholomaus et al (2008) is being cited as an example of where glacier motion is or isn't a simple function of meltwater production.

The paper is referred to as an example of "not a simple function of surface meltwater production". We will move this citation to the next sentence to avoid confusion.

L283-285. I think you can see indications of this even within the short time windows in which data is available each year, with comparable temperature spikes appearing to generate a smaller velocity response later in each measurement period – this is particularly apparent in 2013 and 2019. Also, this sentence would benefit from one or more references – there are plenty to choose from on this topic.

In general, the response of the speed to meltwater input is smaller later in the season as commented by the reviewer. Although we should be careful about the influence of rain (e.g.

the first speed peak in 2013 is due to rain), speed becomes less sensitive to temperature in the later period. We describe this trend with citations, for example, the suggested Davison et al., 2019 paper.

L292. Suggest '···glacier acceleration has been documented···'

We correct the text as suggested.

L295-6. It would perhaps be fair to say that incidences of rain induced acceleration may be increasing in Greenland as the climate warms, but if the authors feel it is truly of 'critical importance' then it would be good to see some justification for this with respect to its contribution to ice discharge.

I agree that the sentence carries too much emphasis. We will revise the text to explain the increasing importance of rain-induced acceleration, but not to mislead the readers.

L311-313. It also looks like temperature is falling at this time, so it may be that the addition of rainfall is partially compensated by a reduction in meltwater.

Temperature decreases by several degrees from 21 to 22 July 2014 (Figure 6b). However, the temperature peak that almost coincided with the rain event on 21 July was still one of the highest during the measurement period. Therefore, we attribute the lack of acceleration to the basal hydrological efficiency rather than the temperature drop.

L316. Rather that saying the subglacial drainage system 'had already developed' (which implies the system can reach a specific 'developed' state beyond which it no longer changes), it might be more correct to say something like 'had already evolved to a more efficient state'.

We agree with the comment. The text will be revised as suggested.

L316-317. As for L311-313, it looks like temperature is dropping at this point too.

I understand that the reviewer is mentioning the temperature drop by several degrees in the evening of 10th July 2017 (Figure 6c), which was approximately 0.5 day after the initiation of the rain event. Considering the swift response of the ice speed to the rain events in 2013 and 2014, the lack of acceleration in 2017 cannot be attributed to the temperature change. The

temperature drop is reflected by progressive deceleration during the rest of the measurement period.

L399. It's also been previously observed (e.g. Cowton et al 2016) that there is a stronger correlation between horizontal velocity and the rate of surface uplift (i.e. the vertical velocity) – have you checked to see whether there is any correlation in this instance?

As far as they are compared with temporal scales of several days to a week, velocity is more correlated with the magnitude of the uplift rather than the uplift rate. Cowton et al. (2016) (as well as Sugiyama et al., 2003) reported a correlation between velocity and uplift rate, but they are compared in shorter temporal resolutions (hours to a day). By the way, I assume the reviewer is referring to another paper below by the same author in the same year.

Cowton, T. et al., 2016, Variability in ice motion at a land-terminating Greenlandic outlet glacier: the role of channelized and distributed drainage systems, J. Glaciol., 62, 451–466, https://doi.org/ 10.1017/jog.2016.36

Sugiyama, S. and Gudmundsson, G. H.: Short-term variations in glacier flow controlled by subglacial water pressure at Lauteraargletscher, Bernese Alps, Switzerland, J. Glaciol., 50, 353–362, https://doi.org/10.3189/172756504781829846,
2004.

L453-4. I've flagged this here as there is a specific reference to mountain glaciers in this sentence, but the comment applies more generally. It's great that earlier work on the hydrology and dynamics of mountain glaciers is cited – this forms the foundation of the topic and it's important that its contribution is recognized. There is however also a substantial body of literature on the hydrology and dynamics of land terminating glaciers in Greenland which builds on this work from mountain glaciers and develops it in a Greenlandic context. This is largely overlooked in the current manuscript, but it would seem appropriate to give this a little more reference, as it is the logical stepping stone between earlier work on alpine glaciers and the current study (and addresses many of the same themes as this manuscript). For a review, see Davison et al (2019).

We acknowledge the reviewer for pointing out the lack of citations. In revision, we refer to the suggested previous studies on land-terminating glaciers in Greenland.

References

Cowton, T., Sole, A., Nienow, P., Slater, D., Wilton, D., & Hanna, E. (2016). Controls on the transport of oceanic heat to Kangerdlugssuaq Glacier, east Greenland. Journal of Glaciology, 62(236), 1167-1180.

Davison, B. J., Sole, A. J., Livingstone, S. J., Cowton, T. R., & Nienow, P. W. (2019). The influence of hydrology on the dynamics of land-terminating sectors of the Greenland ice sheet. Frontiers in Earth Science, 7, 10.

---

## Author Response (AR1)

Reply to Reviewer 1

Thank you very much for reading our manuscript carefully and providing helpful comments. Below (in red), we address the reviewer's comments and explain how we have revised the manuscript. Line numbers are as in the revised version of the manuscript.

This paper uses detailed GPS-based ice velocity records from near the terminus of Bowdoin Glacier, NW Greenland, to explore controls on the short term variability in ice motion along the lowermost 4 km of the glacier. This methodology is quite similar to that employed in previous studies of the dynamics of alpine glaciers and outlet glaciers from the Greenland Ice Sheet, and correspondingly some of the findings are quite familiar from earlier work. However, studies on the short term dynamics of tidewater glaciers remain comparatively uncommon, and this study thus adds value in demonstrating how far our understanding of glacier dynamics applies (and doesn't apply) in this context. The GPS data on which the paper is based is of high quality and the analysis and interpretation seem largely sound and well supported. I have no major concerns but just a few minor comments:

L31. Can you be more quantitative on the relative contribution of these processes?

The relative contributions are described as "In fact, more than 60% of the mass loss from the ice sheet in 1972–2018 was attributed to the acceleration of marine-terminating outlet glaciers, whereas the rest to increasingly negative surface mass balance (Mouginot et al., 2019)" (Line 30–32).

L33. I think it's overstating things to say that this is 'key' (implying it's the single most important factor) – it is one many topics that are important in understanding the current and future mass loss of the ice sheet.

We agree with this comment. "key" is reworded by "crucial" (Line 33).

L249. More so than simply the time taken for water to get to the bed, I think an important consideration here is what controls the timing of the diurnal peak in water pressure. Throughout the middle part of the day when melting is greatest, meltwater input to the glacier will exceed meltwater discharge from the glacier, causing water storage and pressure to increase (the opposite occurs during the remainder of the day). Thus the period of maximum pressure will not likely coincide with maximum melt rates, but rather will occur slightly later

(towards the end of the higher melt period of the day), as you observe (see for example Cowton et al 2016).

The suggested process is included in the text. "This lag implies that the basal sliding was controlled by subglacial water pressure, which was elevated as meltwater input exceeded subglacial water discharge." (Line 257–258)

L255-257. Could snow cover also be a factor here? I assume there was little in the study area at Bowdoin Glacier, but if the study area at Helheim extended 37 km from the terminus, then perhaps remaining snowpack could have played a role in slowing the runoff of meltwater.

Meltwater input from snow-covered upper regions is possible both in Helheim and Bowdoin Glaciers, but delay in runoff due to temporal retention in snowpack is expected to be longer than ~hours. It obscures diurnal ice speed variations, but it is unlikely to cause a "shift" in diurnal peaks. Further, Most of the GPS stations on Helheim Glacier was below 800 m a.s.l. (Stevens et al., 2021), which is well below the equilibrium line altitude. Therefore, the studied area was snow free during the mid and late summer survey period (July and August). Because of the reasons above, we exclude the influence of snowpack from discussion.

L282. Suggest 'The amount and···'

Corrected as suggested (Line 292).

L283. Suggest '···subglacial meltwater input on the basal water pressure···'

Corrected as suggested (Line 293).

L283. It's not totally clear whether Bartholomaus et al (2008) is being cited as an example of where glacier motion is or isn't a simple function of meltwater production.

The paper is referred to as an example of "not a simple function of surface meltwater production". This citation is moved to the next sentence to avoid confusion (Line 293).

L283-285. I think you can see indications of this even within the short time windows in which data is available each year, with comparable temperature spikes appearing to generate a smaller velocity response later in each measurement period – this is particularly apparent in

2013 and 2019. Also, this sentence would benefit from one or more references – there are plenty to choose from on this topic.

In general, the response of the speed to meltwater input is smaller later in the season as commented by the reviewer. Although we should be careful about the influence of rain (e.g. the first speed peak in 2013 is due to rain), speed becomes less sensitive to temperature in the later period. This trend is explained as "In agreement with the hypothesis, the response of the ice speed to temperature rise was generally weaker later in the season" (Line 295–296).

L292. Suggest '⋯glacier acceleration has been documented⋯'

Corrected as suggested (Line 303).

L295-6. It would perhaps be fair to say that incidences of rain induced acceleration may be increasing in Greenland as the climate warms, but if the authors feel it is truly of 'critical importance' then it would be good to see some justification for this with respect to its contribution to ice discharge.

I agree that the sentence carries too much emphasis. The text is revised as "speed up due to rain is important for" (Line 306).

L311-313. It also looks like temperature is falling at this time, so it may be that the addition of rainfall is partially compensated by a reduction in meltwater.

Temperature decreases by several degrees from 21 to 22 July 2014 (Figure 6b). However, the temperature peak that almost coincided with the rain event on 21 July was still one of the highest during the measurement period. Therefore, we attribute the lack of acceleration to the basal hydrological efficiency rather than the temperature drop.

L316. Rather that saying the subglacial drainage system 'had already developed' (which implies the system can reach a specific 'developed' state beyond which it no longer changes), it might be more correct to say something like 'had already evolved to a more efficient state'.

We agree with the comment. The text is revised as suggested (Line 327).

L316-317. As for L311-313, it looks like temperature is dropping at this point too.

I understand that the reviewer is mentioning the temperature drop by several degrees in the evening of 10th July 2017 (Figure 6c), which was approximately 0.5 day after the initiation of the rain event. Considering the swift response of the ice speed to the rain events in 2013 and 2014, the lack of acceleration in 2017 cannot be attributed to the temperature change. The temperature drop is reflected by progressive deceleration during the rest of the measurement period.

L399. It's also been previously observed (e.g. Cowton et al 2016) that there is a stronger correlation between horizontal velocity and the rate of surface uplift (i.e. the vertical velocity) – have you checked to see whether there is any correlation in this instance?

As far as they are compared with temporal scales of several days to a week, velocity is more correlated with the magnitude of the uplift rather than the uplift rate. Cowton et al. (2016) (as well as Sugiyama et al., 2003) reported a correlation between velocity and uplift rate, but they are compared in shorter temporal resolutions (hours to a day). By the way, I assume the reviewer is referring to another paper below by the same author in the same year.

Cowton, T. et al.: Variability in ice motion at a land-terminating Greenlandic outlet glacier: the role of channelized and distributed drainage systems, J. Glaciol., 62, 451–466, https://doi.org/ 10.1017/jog.2016.36, 2016.

Sugiyama, S. and Gudmundsson, G. H.: Short-term variations in glacier flow controlled by subglacial water pressure at Lauteraargletscher, Bernese Alps, Switzerland, J. Glaciol., 50, 353–362, https://doi.org/10.3189/172756504781829846, 2004.

L453-4. I've flagged this here as there is a specific reference to mountain glaciers in this sentence, but the comment applies more generally. It's great that earlier work on the hydrology and dynamics of mountain glaciers is cited – this forms the foundation of the topic and it's important that its contribution is recognized. There is however also a substantial body of literature on the hydrology and dynamics of land terminating glaciers in Greenland which builds on this work from mountain glaciers and develops it in a Greenlandic context. This is largely overlooked in the current manuscript, but it would seem appropriate to give this a little more reference, as it is the logical stepping stone between earlier work on alpine glaciers and the current study (and addresses many of the same themes as this manuscript). For a review, see Davison et al (2019).

Cowton, T., Sole, A., Nienow, P., Slater, D., Wilton, D., & Hanna, E. (2016). Controls on the transport of oceanic heat to Kangerdlugssuaq Glacier, east Greenland. Journal of Glaciology, 62(236), 1167-1180.

Davison, B. J., Sole, A. J., Livingstone, S. J., Cowton, T. R., & Nienow, P. W. (2019). The influence of hydrology on the dynamics of land-terminating sectors of the Greenland ice sheet. Frontiers in Earth Science, 7, 10.

We acknowledge the reviewer for pointing out the lack of citations. This was because we refer to pioneering studies in alpine glaciers to introduce the concept and discussed most relevant studies at tidewater glaciers in Greenland. I agree with the importance of the knowledge from land-terminating outlet glaciers in Greenland, and therefore include additional text and suggested references in Discussion and Conclusion (Line 282, 427–428, 470–471). I would like to note that rain-induced acceleration observed at a land-terminating Greenlandic glacier is also introduced in the manuscript (Line 304–305).

Reply to Reviewer 2

Thank you very much for reviewing our manuscript. Below (in red), we address the review comments and explain how we have revised the manuscript. Line numbers are as in the revised version of the manuscript.

Summary

This manuscript investigates the short-term drivers of speed variability at Bowdoin Glacier, a medium sized outlet glacier in Greenland, using three deployed GPS receivers during six summers from 2013-2019. 3D ice motion is compared to tidal fluctuations and AWS-recorded near-surface air temperature and precipitation. Units were deployed at approximately the same location each year and oriented along flow with one unit nearest the terminus, and the third ~4 km inland. The study finds that glacier speeds responded both diurnally (melt-driven signal) and semi-diurnally (tide-modulated), though the later forcing decayed in influence with increasing distances from the front. Rain-driven acceleration was also detected in some years, though inconsistently across years, which is attributed to the subglacial drainage system evolution and dependency on the state of the subglacial system when precipitation events occur. Tidal influence was strongest near the front, and most pronounced once the terminus is believed to be at or near flotation heights. Most interestingly, this study shows that the 2 inland GPS receivers record uplift during periods (up to multi-day) of acceleration on the order of several centimeters, which is linked to the physical separation of the glacier from the bed as pressurized subglacial drainage systems form.

The manuscript is well written and arranged in a comprehensive and logical structure, with appropriate figures that complement the main results in the text. The study presents important results for understanding drivers and response times of dynamic outlet glaciers and evolving subglacial systems and offers valuable in situ observations that capture processes that occur at higher frequency than can be captured by most remote sensing studies. This manuscript is therefore nearly suitable for publication in TC in its current form, but I find there to be two topics that warrant further analysis/discussion in the main text, mainly: context on the position and phase of the terminus throughout the study period and (2) more figures that describe how key variables (such as lag time and coefficients of temp/speed relationships) vary between years and any trends that were observed. These themes are discussed below in the 'Main', followed by minor comments and requests for clarification.

Main

Terminus change: Some general description of Bowdoin's terminus change are provided in the background, but it would be beneficial to include more information on terminus change during the study period, especially because variations in distance-to-front is found to be an important component on varying responses to tidal impacts between sites 1, 2 and 3. Similarly to the phase of tidal change, the phase of terminus change (whether advancing of retreating), and distance from the nearest GPS receiver (unit 1) each year, and the range in that distance over the seasonal study window, are all important variables that may lend more context to various signals detected during the 2013-2019 period. While I believe the terminus remained relatively stable after 2013 as compared to the large retreat in the preceding years, the interannual variability in when advance/retreat occurs (if any) would still be important to address in this manuscript.

To address the reviewer's concern, we analyzed the variation of the front position over the study years (2013–2019), which is now included as Supplementary Figure S1 (see also below). The front position data were obtained from Zhang et al. (2023) and processed using the box method provided by Lea (2018).

From 2013 to 2019, the glacier front showed seasonal variations with an amplitude of 100–200 m (Fig. S1a). Despite the seasonal position change, the glacier front was situated at similar locations every summer, which were distributed within ~100 m (Fig. S1b–g). During each of the summer measurement periods, the range of the frontal variation was relatively small (typically smaller than 50 m).

Our analysis indicates the change in the glacier position during the measurements was not large as compared to the distance to the GPS sites from the front (~0.5, 2.5 and 4 km). As far as we have investigated the data, there is no significant influence of the front position or its change on the ice speed variations. Further, primary influence of meltwater production on the seasonal ice speed variations was reported in Bowdoin Glacier, whereas no clear relationship was found between the speed change and the front position (Sakakibara and Sugiyama, 2020). Therefore, we show the front position data as Supplementary Figure 1 and describe above in Method (Line124–129), but further analysis and discussion of its influence on the ice speed variations are not performed.

[Figure]

Figure S1. (a) Ice front displacement of Bowdoin Glacier relative to the position in March 2013. (b–g) The displacement during the field measurement periods in 2013–2017 and 2019. The front position data were obtained from Zhang et al. (2023) and processed based on the box method using software provided by Lea (2018).

Lea, J. M.: The Google Earth Engine Digitisation Tool (GEEDiT) and the Margin change Quantification Tool (MaQiT) – simple tools for the rapid mapping and quantification of changing Earth surface margins, Earth Surf. Dynam., 6, 551–561, https://doi.org/10.5194/esurf-6-551-2018, 2018

Sakakibara D., and Sugiyama S.: Seasonal ice-speed variations in 10 marine-terminating outlet glaciers along the coast of Prudhoe Land, northwestern Greenland. J. Glaciol., 66(255), 25–34. https://doi.org/10.1017/jog.2019.81, 2020

Zhang, E., Catania, G., and Trugman, D. T.: AutoTerm: an automated pipeline for glacier terminus extraction using machine learning and a "big data" repository of Greenland glacier termini, The Cryosphere, 17, 3485–3503, https://doi.org/10.5194/tc-17-3485-2023, 2023.

Interannual changes in key variables:

A main strength of this manuscript over previously published studies at this glacier is the extended 2013-2019 study period, which enables the authors to investigate trends and interannual variability in key characteristics of ice flow. While the 'stacking" approach across multiple years was necessary to conduct the fast fourier transform and identify the dominant frequency components, multiyear mean values (such as shown in Figure 4) exclude potentially informative information on how seasonality and characteristics of low have evolved over time. For example, the temperature-max speed lag time of 2 hours was only provided for GPS3, and using a mean result from stacked daily values. It would be useful to understand how this lag compared across years at GPS3, or even compared to lag time at GPS2. Another recommendation on this theme would be to provide corresponding text in the main manuscript that describes the relationships seen in the scatter plots (which include all years superimposed). The plots by themselves are not super informative, and difficult to discern how correlations vary (or remain consistent) across years.

Figure 4 was obtained by subtracting a general trend, stacking data in each year, and taking a mean of the results from six years (Figure S3). As it is seen in Figure S3m, the discussion of the lag between ice speed and temperature peaks is only possible after stacking and taking a mean of available data. It is not possible to discuss seasonal or year-to-year variations based on Figure S3m.

We analyzed the data set from GPS3 as it is not affected by the tide. Because ice motion at GPS2 is influenced by tide (Figure 3), ice speed variations should be discussed with tidal variations as well as temperature.

Further discussion of the scatter plots of ice speed v.s. temperature (Figure 5g–i) are given in Line 288–296. Deviations from the general relationship in 2013 and 2016 are explained by speed-up events. Year-to-year variations are attributed to the efficiency of subglacial drainage efficiency. More detailed discussion for each year is not possible based on our data.

We thank the suggestion and encouragement of the reviewer. However, our six-year data set is just enough for the discussion presented in the manuscript, but not sufficient for detailed analysis of seasonal or year-to-year variations.

Minor

Request for more clarity: Were 2013, 2014 and 2017 the only years where precipitation was recorded during the study period?

The automatic weather station was operated during the field campaigns as described in Line

147. Precipitation was detected in 2013, 2014, 2016, 2017 and 2018 as shown in Figure 2a-f. It is not clear for us why the reviewer misunderstood.

Scatter-plots (for example, in figure 9): Consider using a colormap that avoids very similar colors. It is difficult to discern the years shown in light blue and darker blue (2013 and 2019).

Thank you for pointing out this. The dark blue markers (2013 data) are replaced by dark grey for Figures 5g–i (see belowe) and 9g–i.

[Figure]

Line 80:
Consider replacing "continuous" with "multi-year year series".

I understand that continuous is confusing because the measurements were made only in summer. We revised the text as suggested (multi-year series GPS measurements) (Line 82).

Request for additional citation: I think some of the introductory discussion on subglacial hydrology is light on citations, particularly for more recent work. Should also include citations for discussion on tidal force balance at the calving front. (lines 35-42, and again supporting citations in lines 72-80). There are some important citations used later in the discussion of results on subglacial hydrology that could be incorporated again in the earlier introduction/background.

We include additional references for basal water pressure and tidal influence on ice dynamics (Line 37, 41) and short-term speed variations in Greenland and Alaska (Line 72, 74). Let us note that direct evidence of "basal sliding enhancement due to elevated subglacial pressure" is sparce.

Request for additional clarity: I found some of the detrending methods description confusing. For example, in some places the text uses "detrended" to refer to, what I believe, is simply the time series with the seasonal mean subtracted. In Figure 5, 2019 speeds certainly show a "trend" over the July period, though the mean is zero. However, I assume this detrending approach (where a seasonal acceleration or deceleration may be present) is a different approach used to the "stacking" described, where a mean diurnal speed is computed. Can you please provide more clarity on these methods?

"Detrended ice speed" was obtained by subtracting mean displacement from the positioning data. It is not a simple subtraction of the seasonal mean speed. The details are described in the Method section in Line 139–142. "To investigate the deviation of the ice motion from a general trend, the mean ice motion was subtracted from the positioning data. The mean ice motion was computed by linear regression of the positioning data obtained in each season. The residual speed and vertical displacement were used to discuss ice speed variations and surface uplift." Actually, we do not use the word "detrended" in this method description, which I suppose is the reason for the misunderstanding. We clarify the point by writing "The residual speed and vertical displacement (hereafter referred to as detrended) were used to discuss ice speed variations and surface uplift" (Line 141–142).

Line 170 – how is significant acceleration defined here? Based on a threshold rate of change?

We write, for example, "the glacier significantly accelerated (Line 187)", "significant year-to-year variations (Line 211)" and "semidiurnal variations are less significant (Line 237)" to refer to substantially large changes in ice speed. I understand that the word is used for "statistical significance", but here we use it in place of "substantial", "notable", "considerable". We believe this usage is usual and our texts are not confusing.

Reply to the comment by Ralf Greve

Thank you very much for the comment related to our publication on Bowdoin Glacier (Ralf Greve and Shin Sugiyama are the coauthors of Seddik et al., 2019).

I think the authors should discuss their findings against the results of the modelling study by Seddik et al. (2019). Quoting the abstract:
"Reduction of the basal drag by 10-40% produces speed-ups that agree approximately with the observed range of speed-ups that result from warm weather and precipitation events. In agreement with the observations, tidal forcing and surface speed near the calving front are found to be in anti-phase (high tide corresponds to low speed, and vice versa). However, the amplitude of the semi-diurnal variability is underpredicted by a factor ~3, which is likely related to either inaccuracies in the surface and bedrock topographies or mechanical weakening due to crevassing."
In particular, it would be interesting whether there is any new insight in possible reasons for the underprediction of the amplitude.

Reference:
Seddik, H., R. Greve, D. Sakakibara, S. Tsutaki, M. Minowa and S. Sugiyama. 2019. Response of the flow dynamics of Bowdoin Glacier, northwestern Greenland, to basal lubrication and tidal forcing. Journal of Glaciology 65 (250), 225-238, doi: 10.1017/jog.2018.106.

The amplitude of the tidal ice speed variations modeled by Seddik et al. (2019) was approximately 1/3 of the observation. The modeling was performed at the lowermost GPS site (GPS1). First of all, the model neglected elasticity, which played a role in the tidal modulations according to the tide-speed plots (Figure 7). Second, the power of the semidiurnal signal decays rapidly upglacier. The power at the second GPS (2 km upglacier) is 30% of that at the lowermost GPS, thus it is sensitive to the sampling point. Third, bed elevation map was generated by interpolation of field data (Figure 1b in Seddik et al., 2019). Fourth, The model does not consider fracture of ice, which may be important for ice motion near the calving front. The discrepancy of the modeled results from the observation can be attributed to these processes and details not incorporated in the model. Discussion of this previous study is included in Line 396–401.

---

## Referee Report (RR1)

**Remaining comments:**

**Several of my remaining comments are rebuttals to the initial response to reviewers. I include the original reviewer comment in italicized black text, followed by the author response in red, with my clarification/rebuttal in bolded black text below.**

**On line 30, the authors added a new citation to "The IMBIE Team 2020". There is a newer version of the IMBIE assessment found here:**

*Otosaka, I. N., Shepherd, A., Ivins, E. R., Schlegel, N.-J., Amory, C., van den Broeke, M. R., Horwath, M., Joughin, I., King, M. D., Krinner, G., Nowicki, S., Payne, A. J., Rignot, E., Scambos, T., Simon, K. M., Smith, B. E., Sørensen, L. S., Velicogna, I., Whitehouse, P. L., A, G., Agosta, C., Ahlstrøm, A. P., Blazquez, A., Colgan, W., Engdahl, M. E., Fettweis, X., Forsberg, R., Gallée, H., Gardner, A., Gilbert, L., Gourmelen, N., Groh, A., Gunter, B. C., Harig, C., Helm, V., Khan, S. A., Kittel, C., Konrad, H., Langen, P. L., Lecavalier, B. S., Liang, C.-C., Loomis, B. D., McMillan, M., Melini, D., Mernild, S. H., Mottram, R., Mouginot, J., Nilsson, J., Noël, B., Pattle, M. E., Peltier, W. R., Pie, N., Roca, M., Sasgen, I., Save, H. V., Seo, K.-W., Scheuchl, B., Schrama, E. J. O., Schröder, L., Simonsen, S. B., Slater, T., Spada, G., Sutterley, T. C., Vishwakarma, B. D., van Wessem, J. M., Wiese, D., van der Wal, W., and Wouters, B.: Mass balance of the Greenland and Antarctic ice sheets from 1992 to 2020, Earth Syst. Sci. Data, 15, 1597–1616, https://doi.org/10.5194/essd-15-1597-2023, 2023.*

Original comment:
*It would be useful to understand how this lag compared across years at GPS3, or even compared to lag time at GPS2. Another recommendation on this theme would be to provide corresponding text in the main manuscript that describes the relationships seen in the scatter plots (which include all years superimposed). The plots by themselves are not super informative, and difficult to discern how correlations vary (or remain consistent) across years.*

*"Figure 4 was obtained by subtracting a general trend, stacking data in each year, and taking a mean of the results from six years (Figure S3). As it is seen in Figure S3m, the discussion of the lag between ice speed and temperature peaks is only possible after stacking and taking a mean of available data. It is not possible to discuss seasonal or year-to-year variations based on Figure S3m. .... We thank the suggestion and encouragement of the reviewer. However, our six-year data set is just enough for the discussion presented in the manuscript, but not sufficient for detailed analysis of seasonal or year-to-year variations."*

**I understand that Figure 4 shows the mean of six individual curves from figure S3. These curves are used to determine the mean temporal lag between temperature and speed. Using the curves in Figure S3, can the authors describe how the temporal lag varies inter annually? 2 hours is the mean of 6 six years, but it would be informative to readers to understand if this lag is consistent year-to-year, or exhibits variability in sign and/or magnitude. What is the range in temporal offsets between curves during the six-year study period?**

Original comment:

*Request for more clarity: Were 2013, 2014 and 2017 the only years where precipitation was recorded during the study period?*

*The automatic weather station was operated during the field campaigns as described in Line 147. Precipitation was detected in 2013, 2014, 2016, 2017 and 2018 as shown in Figure 2a-f. It is not clear for us why the reviewer misunderstood.*

**This comment was based on the discussion in Section 5.3 "speed up due to rain" which states that "The glacier responded differently to precipitation events in 2013, 2014, and 2017" and (as well as Figure 6 that shows precipitation during only years 2013, 2014, and 2017). If precipitation also occurred in years 2016 and 2018 (I think the authors meant 2019 instead of 2018 here), then this should also be included in the discussion, even if no coincident speed up was associated with observed precipitation. It's not clear why results from all years (when precip was recorded) were not included.**

Original comment:

*Line 170 – how is significant acceleration defined here? Based on a threshold rate of change?*

*We write, for example, "the glacier significantly accelerated (Line 187)", "significant year-to-year variations (Line 211)" and "semidiurnal variations are less significant (Line 237)" to refer to substantially large changes in ice speed. I understand that the word is used for "statistical significance", but here we use it in place of "substantial", "notable", "considerable". We believe this usage is usual and our texts are not confusing.*

**I find the use of significant here to be confusing and suggest language like "rapid acceleration" or "substantial year to year variability" would be more appropriate in this context.**

Original comment:

Scatter-plots (for example, in figure 9): Consider using a colormap that avoids very similar colors. It is difficult to discern the years shown in light blue and darker blue (2013 and 2019).

*Thank you for pointing out this. The dark blue markers (2013 data) are replaced by dark grey for Figures 5g–i (see belowe) and 9g–i.*

**I am not seeing the updated figures, as described above, reflected in the 'tracked changes' version of the revised manuscript.**

---

## Author Response (AR2)

**Reply to referee comments**

Remaining comments:

Several of my remaining comments are rebuttals to the initial response to reviewers. I include the original reviewer comment in italicized black text, followed by the author response in red, with my clarification/rebuttal in bolded black text below.

We thank the reviewer for carefully reading the revised manuscript. Please see below our reply to your comments in red. Texts in blue are our reply to the first-round review comments.

On line 30, the authors added a new citation to "The IMBIE Team 2020". There is a newer version of the IMBIE assessment found here:
Otosaka, I. N. et al.: Mass balance of the Greenland and Antarctic ice sheets from 1992 to 2020, Earth Syst. Sci. Data, 15, 1597–1616, https://doi.org/10.5194/essd-15-1597-2023, 2023.

This is a helpful suggestion. The citation is replaced.

Original comment:

It would be useful to understand how this lag compared across years at GPS3, or even compared to lag time at GPS2. Another recommendation on this theme would be to provide corresponding text in the main manuscript that describes the relationships seen in the scatter plots (which include all years superimposed). The plots by themselves are not super informative, and difficult to discern how correlations vary (or remain consistent) across years.

"Figure 4 was obtained by subtracting a general trend, stacking data in each year, and taking a mean of the results from six years (Figure S3). As it is seen in Figure S3m, the discussion of the lag between ice speed and temperature peaks is only possible after stacking and taking a mean of available data. It is not possible to discuss seasonal or year-to-year variations based on Figure S3m. .... We thank the suggestion and encouragement of the reviewer. However, our six-year data set is just enough for the discussion presented in the manuscript, but not sufficient for detailed analysis of seasonal or year-to-year variations.

I understand that Figure 4 shows the mean of six individual curves from figure S3. These curves are used to determine the mean temporal lag between temperature and speed. Using the curves in Figure S3, can the authors describe how the temporal lag varies inter annually? 2 hours is the mean of 6 six years, but it would be informative to readers to understand if this lag is consistent year-to-year, or

exhibits variability in sign and/or magnitude. What is the range in temporal offsets between curves during the six-year study period?

We do not include the suggested analysis because we are not able to provide reliable results and useful discussion. Diurnal variations in temperature and speed in each year are not smooth enough to obtain accurate time lag between the peaks (Figure 1). It is difficult to compare the plots to say which year the lag is greater. A quantitative analysis was possible only after averaging curves for six years. We thank the encouragement, but would like not to discuss year-to-year variations.

[Figure]

Figure 1: Diurnal variations in the detrended ice speed (red) and temperature (blue). Each curve was obtained by connecting values averaged in each year.

Original comment:

Request for more clarity: Were 2013, 2014 and 2017 the only years where precipitation was recorded during the study period?

The automatic weather station was operated during the field campaigns as described in Line 147. Precipitation was detected in 2013, 2014, 2016, 2017 and 2018 as shown in Figure 2a-f. It is not clear for us why the reviewer misunderstood.

This comment was based on the discussion in Section 5.3 "speed up due to rain" which states that "The glacier responded differently to precipitation events in 2013, 2014, and 2017" and (as well as Figure 6 that shows precipitation during only years 2013, 2014, and 2017). If precipitation also occurred in years 2016 and 2018 (I think the authors meant 2019 instead of 2018 here), then this should also be included in the discussion, even if no coincident speed up was associated with observed precipitation. It's not clear why results from all years (when precip was recorded) were not included.

Now, I see the point. Thank you for the clarification. And yes, I meant 2013, 2014, 2016, 2017, and "2019". We discuss and show additional plots only for 2013, 2014 and 2017 because these precipitation events are heavier than the others. For weaker precipitation events, ice speed change was not noticeable (Figure 2). To make it clear, we state that our focus is heavy precipitation events with >1 mm per hour (Line 308).

[Figure]

Figure 2: Ice speed at GPS1–3 (red, blue, and green), tidal height (grey), air temperature (black) and precipitation (bar).

Original comment:

Line 170 – how is significant acceleration defined here? Based on a threshold rate of change?

We write, for example, "the glacier significantly accelerated (Line 187)", "significant year-to-year variations (Line 211)" and "semidiurnal variations are less significant (Line 237)" to refer to substantially large changes in ice speed. I understand that the word is used for "statistical significance", but here we use it in place of "substantial", "notable", "considerable". We believe this usage is usual and our texts are not confusing.

I find the use of significant here to be confusing and suggest language like "rapid acceleration" or "substantial year to year variability" would be more appropriate in this context.

We appreciate this help with language. We have corrected "significant" to other words: "large variations" (Line 11), "substantial contribution" (Line 40), "accelerated to a large extent" (Line 187), "substantial year-to-year variations" (Line 211), "rapid and large glacier accelerations" (Line 312), and "substantially accelerated" (Line 454).

Original comment:

Scatter-plots (for example, in figure 9): Consider using a colormap that avoids very similar colors. It is difficult to discern the years shown in light blue and darker blue (2013 and 2019).

Thank you for pointing out this. The dark blue markers (2013 data) are replaced by dark grey for Figures 5g–i (see belowe) and 9g–i.

I am not seeing the updated figures, as described above, reflected in the 'tracked changes' version of the revised manuscript.

Thank you very much for pointing out my fault. The figures (Figs 5 and 9) are replaced.

---

## Author Response (AR3)

November 30, 2024
Shin Sugiyama
Institute of Low Temperature Science
Hokkaido University
Nishi 8, Kita 19, Sapporo 060-0819, Japan
sugishin@lowtem.hokudai.ac.jp

Dear Editor,

Following the instructions, we prepared revised manuscript and figure files as uploaded. Please also consider a graphical abstract, which visually presents our findings. Suggested language corrections are all included. Thank you very much for reading the paper carefully.

Kind regards,

杉山 慎

Shin Sugiyama